# Inverse Probability of Treatment Weighting in 5-Year Quality-of-Life Comparison among Three Surgical Procedures for Hepatocellular Carcinoma

**DOI:** 10.3390/cancers15010252

**Published:** 2022-12-30

**Authors:** Der-Ming Chang, Yu-Fu Chen, Hong-Yaw Chen, Chong-Chi Chiu, King-Teh Lee, Jhi-Joung Wang, Ding-Ping Sun, Hao-Hsien Lee, Yu-Tsz Shiu, I-Te Chen, Hon-Yi Shi

**Affiliations:** 1Division of Digestive Surgery, Department of Surgery, Yuan’s General Hospital, Kaohsiung 80249, Taiwan; 2Department of Clinical Education & Research, Yuan’s General Hospital, Kaohsiung 80249, Taiwan; 3School of Medicine, College of Medicine, I-Shou University, Kaohsiung 82445, Taiwan; 4Department of Medical Education and Research, E-Da Cancer Hospital, Kaohsiung 82445, Taiwan; 5Division of Hepatobiliary Surgery, Department of Surgery, Kaohsiung Medical University Hospital, Kaohsiung 80708, Taiwan; 6Department of Healthcare Administration and Medical Informatics, Kaohsiung Medical University, Kaohsiung 80708, Taiwan; 7Department of Medical Research, Chi Mei Medical Center, Tainan 71004, Taiwan; 8Department of General Surgery, Chi Mei Medical Center, Liouying, Tainan 71004, Taiwan; 9Department of Food Science and Technology, Chia Nan University of Pharmacy and Science, Tainan 71710, Taiwan; 10Department of Business Management, National Sun Yat-sen University, Kaohsiung 80424, Taiwan; 11Department of Medical Research, Kaohsiung Medical University Hospital, Kaohsiung 80708, Taiwan; 12Department of Medical Research, China Medical University Hospital, China Medical University, Taichung 40402, Taiwan

**Keywords:** quality of life, hepatocellular carcinoma, surgery, generalized estimating equations

## Abstract

**Simple Summary:**

In patients who had undergone resection for hepatocellular carcinoma (HCC), scores for most quality-of-life (QOL) subscales were significantly improved by 6 months after resection; QOL scores then remained stable for the rest of the 5-year period of this study. The QOL improvements after laparoscopic surgery or robotic surgery were much larger than improvements after open surgery. Between the 2nd and 5th year postsurgery, however, QOL improvements were larger in robotic surgery patients compared to laparoscopic surgery patients.

**Abstract:**

This prospective longitudinal cohort study analyzed long-term changes in individual subscales of quality-of-life (QOL) measures and explored whether these changes were related to effective QOL predictors after hepatocellular carcinoma (HCC) surgery. All 520 HCC patients in this study had completed QOL surveys before surgery and at 6 months, 2 years, and 5 years after surgery. Generalized estimating equation models were used to compare the 5-year QOL among the three HCC surgical procedures. The QOL was significantly (*p* < 0.05) improved at 6 months after HCC surgery but plateaued at 2–5 years after surgery. In postoperative surveys, the effect size was largest in the nausea and vomiting subscales in patients who had received robotic surgery, and the effect size was smallest in the dyspnea subscale in patients who had received open surgery. It revealed the following explanatory variables for postoperative QOL: surgical procedure type, gender, age, hepatitis C, smoking, tumor stage, postoperative recurrence, and preoperative QOL. The comparisons revealed that, when evaluating QOL after HCC surgery, several factors other than the surgery itself should be considered. The analysis results also implied that postoperative quality of life might depend not only on the success of the surgical procedure, but also on preoperative quality of life.

## 1. Introduction

Hepatic resection is the mainstay curative treatment for hepatocellular carcinoma (HCC) patients. Generally, the three surgical options for HCC (open surgery, laparoscopic surgery, and robotic surgery) are equally effective in terms of medical outcomes [1,2]. Therefore, patients typically select the procedure that optimizes their quality of life (QOL) [1,2,3].

As it obtains comparable long-term clinical outcomes, laparoscopic surgery is considered the standard alternative to conventional open surgery [4,5]. Previous studies have compared outcomes between robotic surgery and laparoscopic surgery and between robotic surgery and open liver resection [4,5]. However, very little surgical outcome data are available to guide surgeons in selecting among these three surgical approaches [6,7]. In Taiwan, the few reports of complications and QOL outcomes after HCC surgery have been limited to robotic surgical procedures [8]. The available literature indicates that robotic surgery provides HCC patients with better clinical outcomes and higher overall satisfaction compared to both laparoscopic surgery and open surgery [6,7,8]. However, none of the three procedures have consistently demonstrated superior outcomes for other aspects of QOL. Until now, most QOL studies of surgical resection of HCC have only evaluated patients at 3 to 6 months after a single postoperative assessment [3]. Additionally, studies of the efficacy of HCC resection have been limited to surgical procedures performed in only one medical institution. Hence, empirical studies using patient-reported QOL are needed to quantify the effectiveness of clinical treatments for HCC. Data obtained in QOL assessments can be used to improve care quality for cancer patients. To our knowledge, this is the first Taiwan study to apply generalized estimating equation (GEE) analysis in a large-scale and long-term prospective cohort study of QOL change after HCC resection. Many studies have used a natural experimental design to examine the impacts of surgical procedures on cancer outcomes [9,10]. However, a major criticism of natural experimental design is that it does not randomly assign patients to different surgical procedures. For example, studies show that HCC patients with specific demographic attributes and clinical attributes tend to prefer a specific surgical procedure, which introduces the potential for selection bias [11]. To our knowledge, the present study is the first to apply inverse probability of treatment weighting (IPTW) in a natural experimental design for a long-term comparison of QOL outcomes among different HCC surgery types. Therefore, the purposes of this study were to couple a natural experimental design with IPTW in a comparison of 5-year QOL among three HCC surgery types and to explore predictors of QOL after HCC resection.

## 2. Materials and Methods

### 2.1. Study Design and Population

The subjects were patients who had undergone surgical resection of HCC performed at one of three southern Taiwan medical centers between January 2012 and December 2015. Inclusion criteria were the following: (1) a histologic or combined radiographic and laboratory diagnosis of HCC; (2) ability to communicate in Chinese or Taiwanese; (3) agreement to participate in a questionnaire survey performed in the hospital ward or by telephone. For accurate assessment of postoperative outcome measures, only patients who had been treated by highly experienced surgeons were analyzed [12]. Thus, the analysis excluded 21 HCC procedures performed by low-volume surgeons (defined as surgeons who had performed three or fewer surgeries within the previous year). Other major exclusion criteria were concurrent malignancy and participation in another QOL study that might have interfered with this study. Figure 1 shows that, during the sample selection period, 676 subjects were eligible to participate. Of these, 156 were excluded because they did not meet the enrollment criteria, declined to participate, or had already died. Therefore, 520 subjects completed the baseline preoperative survey, and 273 subjects completed the preoperative survey and the 6-month, 2-year, and 5-year postoperative surveys. Baseline demographic and clinical data were collected through questionnaire surveys and chart reviews. This study was approved by the Institutional Review Board of Kaohsiung Medical University Hospital (KMUH-IRB-20110002), and written informed consent was obtained from each participant.

### 2.2. Measuring Instruments

The Functional Assessment of Cancer Therapy-Hepatobiliary (FACT-Hep) and the European Organization for Research and Treatment of Cancer (EORTC) QLQ-C30 questionnaires were used to assess QOL. Subscales of the FACT-Hep included three 7-item subscales with score ranges of 0–28 points (physical well-being, PWB; social/family well-being, SWB; functional well-being, FWB), one 6-item subscale with score ranges of 0–24 points (emotional well-being, EWB), and one 18-item subscale with score ranges of 0–72 points (hepatobiliary cancer subscale, HCS) [13]. The PWB, SWB, EWB, and FWB subscales were combined into the FACT-G total subscale. The FACT-G total and HCS subscales were combined into the FACT-Hep total subscale. Higher scores on all subscales of the FACT-Hep were interpreted as better QOL and fewer symptoms.

The EORTC QLQ-C30 is a generic QOL measure and consists of 30 items organized into five areas of functioning subscales (physical functioning, PF; role functioning, RF; emotional functioning, EF; cognitive functioning, CF; social functioning, SF), nine symptom subscales (fatigue, FA; nausea/vomiting, NV; pain, PA; dyspnoea, DY; insomnia, SL; appetite loss, AP; constipation, CO; diarrhea, DI; financial difficulties, FI) and a global health subscale (QL) [14]. Likert scales (from 1 to 7 in the global health subscale and from 1 to 4 in other subscales) are linearly transformed into scores of 0–100 where higher scores indicate better functional status or worse symptomatic problems.

The Chinese versions of the FACT-Hep and EORTC QLQ-C30 have been validated in HCC patients in Taiwan. In all subjects, the QOL measures were administered by the same three research assistants before and after surgery.

The following study characteristics obtained by records reviews and questionnaire interviews were tested as independent variables in this study: surgical procedure, gender, age, education, living with family, marital status, body mass index (BMI), smoking, drinking, Charlson comorbidity index (CCI), hepatitis B, hepatitis C, tumor stage, postoperative average length of stay (ALOS), postoperative 30-day readmission, postoperative recurrence, and preoperative QOL (FACT-Hep and EORTC QLQ-C30 subscales). Co-morbidities were identified by ICD-9-CM codes, which were used to calculate the Deyo–Charlson co-morbidity index [15]. The dependent variables were the postoperative QOL subscales.

### 2.3. Statistical Analysis

The unit of analysis in the present study was each patient with HCC resection. After determining the distribution of observed subjects and the numbers of subjects excluded due to loss, to follow-up, or to refusal to participate at different time points, the baseline data for the study population were first compared by surgery type. Continuous variables were tested for statistical significance by one-way analysis of variance (ANOVA), and categorical variables were tested by Fisher exact analysis.

A propensity score approach was applied to enable the use of IPTW to balance the baseline characteristics of patients among the three surgery types. Each observation was weighted by the inverse of the probability of a patient receiving HCC surgery, given observed confounders identified to the index date. Stabilized inverse probability weights were used to mitigate the influence of very low probabilities estimated by the propensity score model [16]. Weights were derived to obtain estimates representing population-average treatment effects to enable a balanced comparison among the three groups. Treatment was considered the method chosen at the time of consent to this study. Regression models were used to make final inferences, which enabled adjustment for any covariate that remained unbalanced after IPTW [17].

The radar diagrams present the mean score for each QOL subscale in different surgery groups. The GEE approach was performed to explore longitudinal changes in each QOL subscale at different time points in comparison with reference data, i.e., data obtained in surveys performed before surgery and at 6 months, 2 years, and 5 years after surgery. Each QOL subscale was used as a dependent variable as a function of time and effective covariates, which included surgical procedure, gender, age, education, cohabitation with family, marital status, BMI, smoking, drinking, CCI, hepatitis B, hepatitis C, tumor stage, postoperative ALOS, postoperative 30-day readmission, postoperative recurrence, and preoperative QOL. Effective covariates that significantly correlated with each QOL subscale were identified by univariate analysis and entered in the GEE model for multivariate regression analysis. The GEE approach is similar to a repeated-measure ANOVA but is more powerful because it can accommodate incomplete data for individual subjects at one or more assessment points without compromising their remaining data [18]. This approach is also recommended when analyzing incomplete data in longitudinal studies with continuous outcomes.

Effect size (ES) was directly calculated for comparisons of the relative magnitude of change as measured by the QOL measures. That is, ES was calculated as the difference between the mean scores for two time intervals divided by the standard deviation in the score for the previous (or former) time-interval [19]. Using this method of standardizing the extent of change measured by an instrument enabled comparisons between the QOL measures. An ES of 1.0 is equivalent to a change of one standard deviation (SD) in the sample. Effect sizes of 0.2, 0.5, and 0.8 are typically considered small, medium, and large changes, respectively.

As no studies have quantified the uncertainty in estimated responsiveness, the precision of reported ES values remains unknown, and the statistical results of different studies are difficult to compare across different populations or activity measures. When repeated measures are used, matters are further complicated by dependent observations of the same patient. To address these issues, differences in ES and associated 95% confidence intervals are calculated using bias-corrected and accelerated bootstrapping with 1000 replications [20]. For the statistical analyses in this study, Stata, version 12.0 (StataCorp, College Station, TX, USA) was used to perform GEE in XTGEE. All tests were two-sided, and *p* values less than 0.05 were considered statistically significant.

## 3. Results

### 3.1. Study Population

Table 1 shows the characteristics of the 520 HCC surgery patients (362 with open surgery, 112 with laparoscopic surgery, and 46 with robotic surgery) in this study. The average age was 65.94 years, and 70.4% were male. After adjustment by IPTW, all covariates were well balanced. In all QOL subscales, subjects who continuously participated in the study throughout the 5 years did not significantly differ from those who died or dropped out during the observation period of the study (data not shown).

### 3.2. Longitudinal Changes in QOL

Figure 2 and Figure 3 show the mean scores for the FACT-Hep and EORTC QLQ-C30 subscales, respectively, from the preoperative survey to the 6-month postoperative survey. In all HCC surgery patients, QOL improvements after laparoscopic surgery or robotic surgery were much larger than QOL improvements after open surgery. However, QOL improvements after robotic surgery were larger than those after laparoscopic surgery. The radar diagram clearly distinguishes QOL outcomes among the different surgery types at different time points.

### 3.3. Difference and 95% Confidence Interval (CI) in ES by Using Bootstrapping Method

We compared the difference and 95% CI in ES values for all QOL subscales among the three surgery types at different time points (see Appendix A). A difference is considered statistically significant at the 0.05 level if the 95% CI does not include zero. Between the 6-month and preoperative surveys, both laparoscopic surgery patients and robotic surgery patients compared to open surgery patients had significantly larger absolute differences in ES values for the QOL subscales, but robotic surgery patients compared to laparoscopic surgery patients had significantly smaller absolute differences in ES values for the QOL subscales (*p* < 0.05). Furthermore, both laparoscopic surgery patients and robotic surgery patients compared to open surgery patients showed significantly larger absolute differences in ES values for the QOL subscales between the 5-year and 2-year postoperative surveys (*p* < 0.05), but the absolute differences were smaller than the absolute differences between the 2-year and 6-month postoperative surveys. Additionally, robotic surgery patients compared to laparoscopic surgery patients revealed significantly larger absolute differences in ES values for the QOL subscales during the same time period (*p* < 0.05).

### 3.4. Multivariate Analysis

The GEE models of effective QOL predictors in the HCC surgery patients in this study indicated that each time point was significantly related to the QOL subscales throughout the 5 years (*p* < 0.05) (Table 2 and Table 3). Compared to open surgery patients, laparoscopic surgery patients had significantly higher scores for FACT-Hep SWB and EORTC-QLQ-C30 EF after controlling for related variables; compared to open surgery patients, robotic surgery patients also had significantly higher scores for FACT-G and EORTC-QLQ-C30 CF after controlling for related variables (*p* < 0.05). Compared to open surgery patients, however, robotic surgery patients had significantly lower symptom subscale scores for EORTC-QLQ-C30 PA, SL, AP, and DI (*p* < 0.05). The QOL subscale scores had significantly negative associations with female gender, advanced age, hepatitis C, smoking, high tumor stage, and postoperative recurrence (*p* < 0.05). Additionally, all preoperative QOL subscale scores had significant associations with each subscale score of the FACT-Hep and EORTC-QLQ-C30 throughout the 5-year follow-up surveys (*p* < 0.05).

## 4. Discussion

Comparisons of QOL improvements between different time points indicated that the FACT-Hep and EORTC QLQ-C30 subscale scores for HCC patients were significantly improved by 6 months after resection (*p* < 0.05) and then remained stable for the rest of the 5-year period of the study. The QOL improvements at 6 months postsurgery were also much larger for both laparoscopic surgery and robotic surgery compared to open surgery. Between the 2nd and the 5th years postoperatively, however, QOL was higher in the robotic surgery patients compared to the laparoscopic surgery patients, which is consistent with the literature [3,4,21]. This prospective study of real-world registry data from multiple institutions in Taiwan over a 5-year period found that surgical procedures, gender, age, hepatitis C, smoking, tumor stage, recurrence after surgery, and preoperative QOL subscale scores were significantly associated with QOL subscale scores after hepatocellular carcinoma surgery

This study has several strengths. To our knowledge, this is the first population-based prospective cohort study designed to assess changing trends in postoperative QOL subscale scores in HCC patients after surgery and to evaluate predictors of these scores. Notably, however, the results of this study were obtained in a specific setting: three large tertiary academic hospitals in Taiwan. Nevertheless, the subjects were a representative sample of HCC patients who had received resection performed by high-volume surgeons in Taiwan [22]. Additionally, we used the IPTW method to obtain comparison groups that were balanced in all baseline characteristics.

Any observational study is potentially subject to measured and unmeasured confounding effects. However, this study used IPTW to balance the baseline patient attributes and clinical attributes in a pragmatic nonrandomized cluster design. The results obtained in this approach mimicked those obtained in a randomized clinical trial for a primary outcome and provided a robust sample size while avoiding the individual selection bias present in registries created for observational studies [16]. Additionally, our novel use of IPTW also provided a methodology for maximizing accuracy in comparisons of the three surgical procedures in terms of factors that were potentially related to confounding variables. The results suggest that our comparisons were well balanced and improve our confidence that the estimates were appropriately adjusted for potential confounding variables measured in previous studies and included in our analyses.

A systematic review by Muzellec et al. found that all instruments conventionally used to measure QOL in cancer patients have had adequate development and validation processes, including EORTC QLQ-C30, EORTC QLQ-HCC18, FACT-Hep, FACT Hepatobiliary Symptom Index (FHSI), and Quality of Life-liver cancer (QOL-LC) [2]. The EORTC QLQ-C30 and FACT-Hep are well-developed measures that have been tested extensively in patients with HCC [13,14]. Luckett et al. further compared the EORTC QLQ and FACT cancer-specific measures for the purpose of informing the choice between them [23]. They concluded that, while further psychometric evidence is still needed, important differences in the subscale structures and social domains of the two measures should inform the choice of which measure to use in a particular study.

However, they also recommended that individual items on the FACT should be reviewed to ensure that symptoms and issues included within each subscale do not produce bias in the context of specific research objectives in a given study. Thus, the current study used the EORTC QLQ-C30 and FACT-Hep measures to survey patient-reported QOL.

The magnitude of improvement in QOL subscales was larger between the preoperative and postoperative 6-month surveys than between the postoperative 2-year and 5-year surveys. In all HCC surgery patients, however, the ES for EORTC QLQ-C30 NV was larger in comparisons between the preoperative and 6-month postoperative surveys than in comparisons between time periods. The magnitude of improvement in the RF and CF subscale scores was smaller than that in other subscale scores, probably because low QOL substantially reduced RF. After undergoing surgery that eliminated their physical and emotional problems, and after completing adjunct treatment, the patients substantially improved in functions that had been previously limited by nausea and vomiting [24]. Thus, alleviating nausea and vomiting can enhance functioning in other subscales of health (e.g., social functioning) and ultimately enhance overall QOL. This improvement may explain why, for most QOL subscales, the absolute ES for the period between the preoperative survey and the postoperative 5-year survey remained positive, whereas the ES for EORTC QLQ-C30 NV was larger in the period between the preoperative survey and postoperative 6-month survey than between the postoperative 2-year survey and postoperative 5-year survey.

For almost all FACT-Hep and EORTC QLQ-C30 subscales related to HCC cancer-symptom functioning (including EF, FA, NV, DY, SL, AP, CO, DI, and FI), the improvement was smaller in the 5-year survey than in both the 6-month survey and 2-year survey. The only exception was the future perspective subscale. Emotional dysfunction in HCC patients may result from adjuvant systemic therapy after surgery [25,26]. In the current study, physical and emotional functioning tended to be highest in patients with short follow-up intervals, whereas overall well-being tended to be highest in patients with long follow-up intervals, which is in line with other studies [2,3,25,26]. One explanation for these observations is the use of coping mechanisms by patients to deal with the stress caused by adverse diagnoses and the potentially life-threatening effects of primary therapy. Gradual decreases in psychological stress may also explain consistent improvements in function symptoms [27,28].

Our findings demonstrated that robotic surgery outperforms laparoscopic surgery and open surgery in terms of PF, RF, EF, CF, SF, and symptoms. Compared with the laparoscopic surgery and open surgery groups, however, the robotic surgery groups revealed significantly larger subjective improvements in PF, RF, EF, CF, SF, and symptoms, which is consistent with previous studies [2,3,25]. A possible explanation for relatively larger improvements in the robotic surgery group is that patients in this group tended to have better socioeconomic status and tended to have a higher education level compared to those in the laparoscopic surgery and open surgery groups. Thus, the larger improvements in the robotic surgery group may have at least partially resulted from factors such as their greater capability to acquire and comprehend health-related information as well as the financial means to optimize their post-surgery recovery conditions.

Finally, the most important predictors of postoperative QOL subscale scores throughout the 5-year study were preoperative QOL subscale scores, which is consistent with reports that preoperative QOL subscale scores are the best predictors of postoperative QOL [29,30]. Therefore, effective counseling is essential for apprising patients of expected postoperative impairments. If QOL outcomes are considered benchmarks, then preoperative QOL subscale scores, which are accurate predictors of postoperative outcome, are crucial.

For further validation of the significant association observed between risk factors and postoperative QOL after HCC surgery, Table 4 lists selected studies that have identified QOL trends after HCC surgery and risk factors for decreased QOL [30,31,32,33,34]. Notably, our cohort study had a larger population and a longer duration of longitudinal analysis compared with the selected studies. As in these previous works, our study demonstrated that QOL was significantly (*p* < 0.05) improved at 6 months after HCC surgery and then plateaued at 2–5 years after surgery. Our study also revealed that QOL after HCC surgery is significantly associated with surgery type, gender, age, hepatitis C, smoking, tumor stage, postoperative recurrence, and preoperative QOL (*p* < 0.05).

### Study Limitations

Although all research questions were adequately and satisfactorily addressed, some limitations of this study are noted. First, this study collected data from patients who had received HCC resection under the supervision of a surgeon in one of three medical centers. Each surgeon had performed the highest volume of HCC resection in their respective hospital. This sample selection procedure was intended to limit the effect of the learning curve on QOL outcomes. As this study focused on high-volume surgeons in three different institutions, the results obtained are more representative of all patients with HCC resection compared to one that analyzes patients treated by a single surgeon. However, a notable limitation is that, in this prospective cohort study, the first patient was enrolled in 2012. Therefore, depending on their inclusion dates, the duration of follow-up differed among patients, which may have caused selection bias.

## 5. Conclusions

In conclusion, most QOL subscale scores for patients with HCC resection were significantly (*p* < 0.05) improved by 6 months postsurgery and then remained stable for the rest of the 5-year study period. Additionally, the QOL improvements at 6 months postsurgery were much larger for laparoscopic surgery and robotic surgery compared to open surgery. Between 2 years and 5 years postsurgery, however, QOL improvement was larger for robotic surgery than for laparoscopic surgery. Nevertheless, a forecast of QOL after HCC resection must consider several factors other than the surgery type itself. All predictors analyzed in this study could be addressed in preoperative and postoperative consultations so that candidates for HCC resection and patients who have already completed HCC resection are adequately educated in the expected course of recovery and expected QOL outcomes. Patients should also be advised that their postoperative quality of life might depend not only on the success of their operations, but also on their preoperative quality of life.

## Figures and Tables

**Figure 1 cancers-15-00252-f001:**
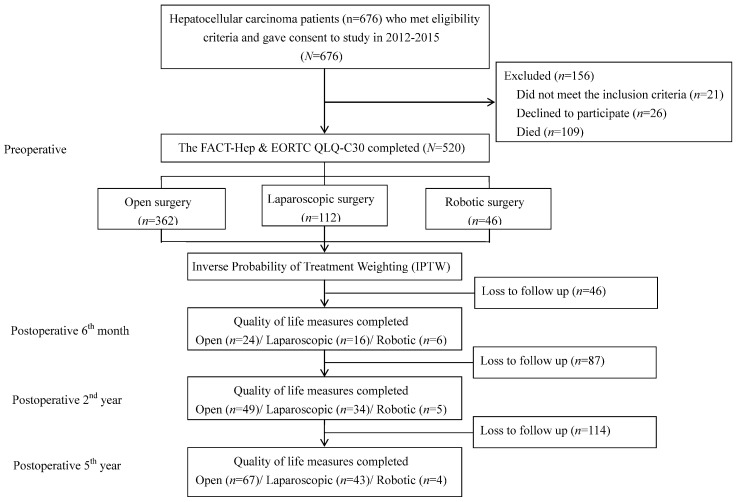
Flowchart of data collection.

**Figure 2 cancers-15-00252-f002:**
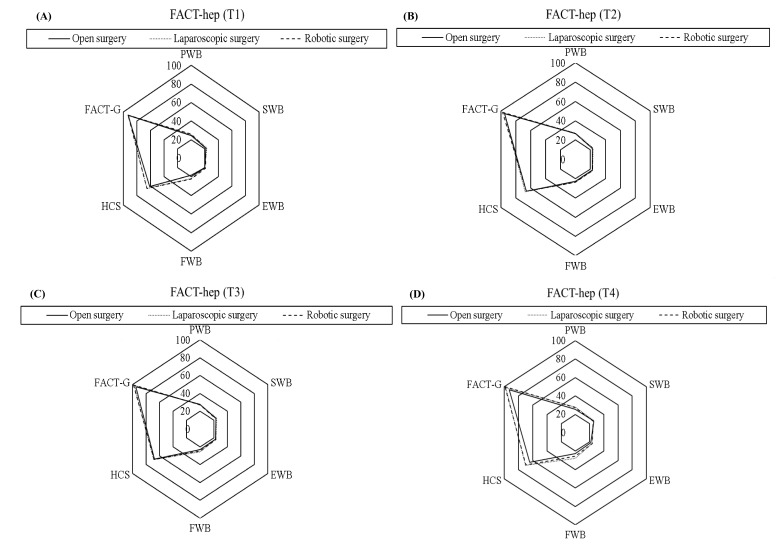
The radar diagram showed the longitudinal changes in the Functional Assessment of Cancer Therapy- Hepatobiliary (FACT-Hep) in hepatocellular carcinoma patients compared among three surgery types at different time points. (**A**) at preoperative (baseline); (**B**) at 6th month postoperative; (**C**) at 2nd year postoperative; (**D**) at 5th year postoperative. PWB, physical well-being; SWB, social/family well-being; EWB, emotional well-being; FWB, functional well-being; HCS, hepatobiliary cancer subscale. T1: preoperative; T2: postoperative 6-month; T3: postoperative 2-year; T4: postoperative 5-year survey.

**Figure 3 cancers-15-00252-f003:**
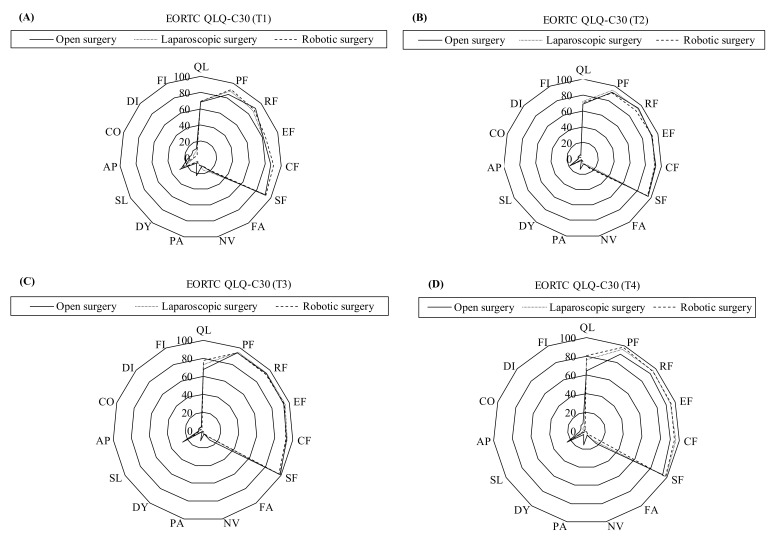
The radar diagram showed the longitudinal changes in the European Organisation for Research and Treatment of Cancer Quality of Life Questionnaire Core 30 (EORTC-QLQ-C30) in hepatocellular carcinoma patients compared among three surgery types at different time points. (**A**) at preoperative (baseline); (**B**) at 6th month postoperative; (**C**) at 2nd year postoperative; (**D**) at 5th year postoperative. QL, global health status; PF, physical functioning; RF, role functioning; EF, emotional functioning; CF, cognitive functioning; SF, social functioning; FA, fatigue; NV, nausea and vomiting; PA, pain; DY, dyspnea; SL, insomnia; AP, appetite loss; CO, constipation; DI, diarrhea; FI, financial difficulties. T1: preoperative; T2: postoperative 6-month; T3: postoperative 2-year; T4: postoperative 5-year survey.

**Table 1 cancers-15-00252-t001:** Hepatocellular carcinoma patient characteristics compared among the three surgical procedures before and after matching by inverse probability of treatment weighting (IPTW).

Variables		Before IPTW Matching				After IPTW Matching			
		Open Surgery (*n* = 362)	Laparoscopic Surgery (*n* = 112)	Robotic surgery (*n* = 46)	*p* Value	Open Surgery	Laparoscopic Surgery	Robotic Surgery	*p* Value
Gender	Male	253 (69.9%)	79 (70.5%)	34 (73.9%)	0.853	253 (69.9%)	75 (67.4%)	33 (72.7%)	0.189
	Female	109 (30.1%)	33 (29.5%)	12 (26.1%)		109 (30.1%)	37 (32.6%)	13 (27.3%)	
Age, years		66.1 ± 10.31	64.07 ± 11.24	69.22 ± 11.82	0.020	66.96 ± 9.03	66.95 ± 9.12	66.95 ± 11.31	1.000
Living with family members	No	40 (11%)	12 (10.7%)	12 (26.1%)	0.012	45 (12.5%)	14 (12.8%)	7 (15.0%)	0.459
	Yes	322 (89%)	100 (89.3%)	34 (73.9%)		317 (87.5%)	98 (87.2%)	39 (85.0%)	
Marital status	Single/other	48 (13.3%)	14 (12.5%)	7 (15.2%)	0.901	48 (13.3%)	15 (13.3%)	8 (16.3%)	0.286
	Married	314 (86.7%)	98 (87.5%)	39 (84.8%)		314 (86.7%)	97 (86.7%)	38 (83.7%)	
Body mass index, kg/m^2^		24.7 ± 3.66	25.11 ± 3.23	23.53 ± 4.36	0.047	24.7 ± 3.66	25.11 ± 3.23	23.53 ± 4.36	0.147
Smoker	No	289 (80.3%)	85 (75.9%)	34 (73.9%)	0.429	286 (79.2%)	90 (80.6%)	35 (75.9%)	0.179
	Yes	71 (19.7%)	27 (24.1%)	12 (26.1%)		76 (20.8%)	22 (19.4%)	11 (24.1%)	
Drinker	No	275 (76.4%)	81 (72.3%)	32 (69.6%)	0.468	271 (74.8%)	84 (74.8%)	33 (72.7%)	0.667
	Yes	85 (23.6%)	31 (27.7%)	14 (30.4%)		91 (25.2%)	28 (25.2%)	13 (27.3%)	
Charlson comorbidity index	0	104 (28.7%)	31 (27.7%)	18 (39.1%)	0.719	108 (29.9%)	29 (26.1%)	16 (33.9%)	0.739
	1	145 (40.1%)	46 (41.1%)	15 (32.6%)		151 (41.6%)	47 (41.6%)	14 (30.9%)	
	2	42 (11.6%)	14 (12.5%)	7 (15.2%)		37 (10.3%)	14 (12.4%)	8 (18.0%)	
	≧3	71 (19.6%)	21 (18.8%)	6 (13%)		66 (18.2%)	22 (19.9%)	8 (17.2%)	
Hepatitis B	No	248 (68.5%)	73 (65.2%)	36 (78.3%)	0.272	249 (68.7%)	80 (71.7%)	34 (73.5%)	0.237
	Yes	114 (31.5%)	39 (34.8%)	10 (21.7%)		113 (31.3%)	32 (28.3%)	12 (26.5%)	
Hepatitis C	No	292 (80.7%)	94 (83.9%)	35 (76.1%)	0.504	292 (80.8%)	90 (80.4%)	37 (80.4%)	0.943
	Yes	70 (19.3%)	18 (16.1%)	11 (23.9%)		70 (19.2%)	22 (19.6%)	9 (19.6%)	
Tumor stage	I	177 (48.9%)	52 (46.4%)	21 (45.7%)	0.303	174 (48.2%)	56 (49.6%)	19 (41.9%)	0.109
	II	103 (28.5%)	39 (34.8%)	13 (28.3%)		109 (30.2%)	33 (29.7%)	16 (34.3%)	
	III	51 (14.1%)	13 (11.6%)	9 (19.6%)		49 (13.5%)	16 (14.2%)	8 (17.9%)	
	≧IV	31 (8.6%)	8 (7.1%)	3 (6.5%)		30 (8.1%)	7 (6.5%)	3 (6.5%)	
Postoperative length of stay, days		11.52 ± 7.43	10.08 ± 9.56	8.54 ± 3.9	0.023	10.31 ± 3.22	9.95 ± 3.48	9.93 ± 3.88	0.161
30-day readmission	No	319 (88.1%)	98 (87.5%)	38 (82.6%)	0.567	317 (87.6%)	100 (89.0%)	40 (87.0%)	0.629
	Yes	43 (11.9%)	14 (12.5%)	8 (17.4%)		45 (12.4%)	12 (11.0%)	6 (13.0%)	
Postoperative recurrence	No	169 (46.7%)	53 (47.3%)	26 (56.5%)	0.451	174 (48.2%)	58 (52.0%)	25 (53.7%)	0.211
	Yes	193 (53.3%)	59 (52.7%)	20 (43.5%)		188 (51.8%)	54 (48.0%)	21 (46.3%)	

**Table 2 cancers-15-00252-t002:** Predictors of each Functional Assessment of Cancer Therapy-Hepatobiliary (FACT-Hep) subscale score after hepatocellular carcinoma surgery over a 5-year period.

Variables ^†^		PWB	SWB	EWB	FWB	HCS	FACT_G	Total
		Coef.	Coef.	Coef.	Coef.	Coef.	Coef.	Coef.
Surgical procedure	Laparoscopic	0.03	0.50 *	0.28	−0.04	0.21	0.90	0.15
	Robotic	0.08	0.46	0.14	0.88	0.57	2.06 *	−1.40
Gender	Female	−0.09	−0.44	−0.13	−1.02 *	−0.16	−0.91	−1.84 *
Age, years		0.01	−0.01	0.01	−0.01	−0.02	0.01	0.01
Living with family	Yes	0.45	−0.50	−0.37	−0.74	0.05	−1.00	−0.99
Marital status	Married	−0.16	−0.03	−0.36	−0.89	−0.11	−1.16	−1.44
Body mass index, kg/m^2^		−0.02	−0.04	0.03	−0.03	−0.01	0.05	0.06
Hepatitis B	Yes	−0.34 *	0.12	0.35 *	0.13	−0.16	0.01	−0.09
Hepatitis C	Yes	0.19	0.21	0.06	0.18	0.52	0.55	0.60
Smoker	Yes	−0.13	0.02	−0.18	−0.30	−0.74 *	0.58	−1.61 *
Drinker	Yes	0.20	0.13	0.43 *	0.04	0.27	0.41	0.22
Charlson comorbidity index	1	0.14	0.19	0.05	−0.28	0.31	−0.14	−1.12
	2	−0.24	0.13	0.48 *	−0.32	−0.01	−0.28	−0.69
	≧3	0.04	0.03	−0.02	−0.77	−0.12	−0.64	−1.82
Postoperative length of stay, days		0.01	0.01	−0.02	0.01	0.01	−0.03	0.04
Postoperative 30-day readmission	Yes	0.02	0.87 **	0.44 **	1.05 *	0.13	2.29 **	1.90 *
Tumor stage	II	0.08	−0.27	0.01	−0.43	0.15	−0.59	0.42
	III	0.35	0.06	0.23	−0.68	0.07	0.39	−1.23
	IV	0.14	−0.11	0.30	0.15	0.81 *	0.85	0.86 *
Recurrence after surgery	Yes	−0.13	0.39	0.12	0.38	−0.03	0.60	0.71
Preoperative quality of life		0.31 ***	0.42 ***	0.28 ***	0.43 ***	0.29 ***	0.41 ***	0.38 ***

PWB, physical well-being; SWB, social/family well-being; EWB, emotional well-being; FWB, functional well-being; HCS, hepatobiliary cancer subscale. ^†^ Reference group: surgical procedure (open surgery), gender (male), living with family (no), marital status (single/other), hepatitis B (no), hepatitis C (no), smoker (no), drinker (no), Charlson comorbidity index (0), postoperative 30-day readmission (no), tumor stage (I), recurrence after surgery (no). * *p* < 0.05, ** *p* < 0.01, *** *p* < 0.001.

**Table 3 cancers-15-00252-t003:** Predictors of each European Organization for Research and Treatment of Cancer Quality of Life Questionnaire Core 30 (EORTC-QLQ-C30) subscale after hepatocellular carcinoma surgery over a 5-year period.

Variables ^†.^		QL	PF	RF	EF	CF	SF	FA	NV	PA	DY	SL	AP	CO	DI	FI
		Coef.	Coef.	Coef.	Coef.	Coef.	Coef.	Coef.	Coef.	Coef.	Coef.	Coef.	Coef.	Coef.	Coef.	Coef.
Surgical procedure	Laparoscopic	1.02	0.35	0.99	2.14 **	−0.04	1.06	0.72	−0.52	−0.69	0.15	−0.52	−0.51	−0.26	−0.91	1.02
	Robotic	−1.15	0.95	−1.91	−0.27	3.15 *	0.05	0.44	0.55	−1.92 *	0.55	−7.18 **	−2.17 *	−0.66	−2.14 *	0.97
Gender	Female	−3.09 **	−1.12	−0.83	−1.36 *	0.26	−0.02	−0.58	−0.95	−0.95	−0.77	1.74	0.68	0.47	−0.23	1.05
Age, years		−0.09	−0.01	0.10	0.16 *	0.11 *	−0.05	−0.55	−0.02	0.04	0.02	−0.05	−0.05	0.09 *	0.05	0.03
Living with family	Yes	−2.01	−0.67	−4.06 *	−0.94	−2.23	−2.25	2.12	−0.44	0.09	0.56	8.79 *	−1.29	−3.89	1.29	−1.20
Marital status	Married	−0.87	−0.20	−0.43	−1.82	−1.66	−0.93	−1.22 *	0.08	2.33 ***	0.95	0.36	−0.33	−0.71	0.17	2.75 *
Body mass index, kg/m^2^		−0.22	0.18	0.19	−0.02	0.06	−0.05	0.18	0.07	−0.06	0.07	−0.17	−0.01	−0.11 *	0.24	−0.14 *
Hepatitis B	Yes	−0.40	−0.35	0.09	1.03	1.19	0.04	−0.09	0.14	0.45	−0.14	−1.24	−1.10	1.17	0.74	−1.48
Hepatitis C	Yes	0.67	1.27	2.12	−0.53	0.66	1.62 *	1.91 **	−0.41	−0.81	−0.25	−0.96	−1.94 *	−0.58	−1.13	−0.58
Smoker	Yes	−1.64	−1.25	−2.95 *	0.18	1.92	−0.63	−0.98 *	−0.11	0.05	1.67 *	0.46	0.23	−0.39	0.93	1.56
Drinker	Yes	0.90	1.58 *	3.22 **	1.45	1.39 *	0.28	0.38	0.53	−0.07	0.09	−0.12	−0.59	−0.68	1.94	−0.56
Charlson comorbidity index	1	−0.77	0.89 *	−1.38	0.26	−0.09	−0.58	−0.09	0.17	−1.35	−1.26	2.24	−0.46	0.82	−0.69	0.27
	2	−0.02	0.43	−0.33	13.0	−2.83	−1.92	0.51	0.64	0.22	−0.26	0.58	−0.31	0.31	0.77	0.95
	≧3	1.17	−0.65	−4.22 *	−0.65	−0.65	−1.49	−0.32	−0.78	−1.13	−1.78 *	1.65	0.52	0.87	−0.52	0.04
Postoperative length of stay, days		0.07	0.09	0.13	−0.01	−0.12	0.04	0.03	0.01	0.02	−0.02	−0.08	−0.07	0.06	−0.02	0.01
Postoperative 30-day readmission	Yes	−0.47	0.01	−2.03	1.72 *	1.40 *	0.64	−0.20	−0.89	0.26	−0.71	−0.28	0.66	−0.56	−0.69	−0.44
Tumor stage	II	−2.25	2.62 **	1.79 *	0.34	2.33	−0.26	0.09	−0.20	−0.10	0.42	−3.44	0.98	−0.38	1.43	0.24
	III	−1.12	1.80 *	2.32 *	1.05	0.84	1.07	0.95	−1.54 *	−1.43 *	0.51	1.22 *	0.78	1.47	−0.03	−1.85
	IV	−2.73	1.33	1.27	0.64	3.04 *	2.40 **	0.95	−2.59 **	0.17	−0.88	−5.10	−1.35	1.98	−1.38	−0.75
Postoperative recurrence	Yes	0.58	−0.53	−0.52	0.08	2.26 **	0.01	−0.90	1.14 *	−0.20	−0.54	1.63 *	−0.60	−0.37	0.38	3.19 *
Preoperative quality of life		0.34 ***	0.54 ***	0.37 ***	0.35 ***	0.44 ***	0.26 ***	0.30 ***	0.31 ***	0.22 ***	0.37 ***	0.36 ***	0.31 ***	0.38 ***	0.25 ***	0.37 ***

QL, global health status; PF, physical functioning; RF, role functioning; EF, emotional functioning; CF, cognitive functioning; SF, social functioning; FA, fatigue; NV, nausea and vomiting; PA, pain; DY, dyspnea; SL, insomnia; AP, appetite loss; CO, constipation; DI, diarrhea; FI, financial difficulties. ^†^ Reference group: surgical procedure (open surgery), gender (male), living with family (no), marital status (single/others), hepatitis B (no), hepatitis C (no), smoker (no), drinker (no), Charlson comorbidity index (0), postoperative 30-day readmission (no), tumor stage (I), recurrence after surgery (no). * *p* < 0.05, ** *p* < 0.01, *** *p* < 0.001.

**Table 4 cancers-15-00252-t004:** Selected studies of changing trends and risk factors for decreased quality of life (QOL) after hepatocellular carcinoma (HCC) surgery.

Authors (Country)	No. of Subjects	Measures	Findings
Chang et al., 2022 (Taiwan) [Present study]	520 patients with HCC surgery	FACT-Hep and EORTC-QLQ-C30	The QOL was significantly (*p* < 0.05) improved at 6 months after HCC surgery and plateaued at 2–5 years after surgery. In postoperative surveys, effect size was largest in the nausea and vomiting subscales for patients who had received robotic surgery and lowest in the dyspnea subscale for those who had received open surgery. Type of surgical procedure, gender, age, hepatitis C, smoking, tumor stage, postoperative recurrence, and preoperative QOL were statistically associated with postoperative QOL (*p* < 0.05).
Chiu et al., 2019 (Taiwan) [30]	369 patients with hepatic resection for HCC	FACT-Hep and SF-36	Predictors of low postoperative QOL included low preoperative QOL score, advanced age, high education level, and high body mass index (*p* < 0.05).
Studer et al., 2018 (Switzerland) [31]	188 patients (130 with malignant and 58 with benign tumors) requiring major liver resection	EORTC QLQ-C30 and EORTC QLQ-LMC21	All patients showed improved global health status at 3, 6, and 12 months postsurgery. Patients with benign disease had better physical function scores (*p* = 0⋅011, *p* = 0⋅025, and *p* = 0⋅041) and lower fatigue scores (*p* = 0⋅001, *p* = 0⋅002, and *p* = 0⋅002) at 3, 6, and 12 months postsurgery compared to those with malignant disease.
Giuliani et al., 2014 (Italy) [32]	75 patients undergoing liver resection	SF-36	Length of stay (4.7 vs. 8.2 days, *p* = 0.0002) and reprisal of oral intake (II post-op vs. III post-op, *p* = 0.02) were lower in the laparoscopic group compared to the open group. QOL was significantly better in the laparoscopic group compared to the open group at postoperative 1-year (*p* < 0.001).
Toro et al., 2012 (Italy) [33]	Fifty-one HCC patients treated with hepatic resection	FACT-Hep	Compared to open surgery, hepatic resection achieves better QOL at 24 months postsurgery.
Martin, et al., 2007 (USA) [34]	32 patients with malignant liver resection	FACT-Hep, FACT-FHSI-8, EORTC QLQ-C30, Profile of Mood States, EORTC QLQ-Pan26, and Global Rating Scale	After major hepatectomy, FACT-physical and functional scores were significantly decreased at the first postoperative visit and at the 6-week postoperative visit (*p* = 0.04) but returned to baseline at the 3-month postoperative visit. For minor hepatectomy, the nadir for most QOL scores occurred at the first postoperative visit with a return to baseline at the 6-week postoperative visit.

## Data Availability

Data and study materials can be made available for non-commercial use upon reasonable request to the corresponding author.

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
