# Peer review of "Inverse Probability of Treatment Weighting in 5-Year Quality-of-Life Comparison among Three Surgical Procedures for Hepatocellular Carcinoma"

_cancers, 2022, doi:10.3390/cancers15010252_

Round 1

Reviewer 1 Report (Previous Reviewer 1)

The article has been significantly improved as per the suggestions made and it will be of great help to optimize the quality of Life of patients undergone surgery for Hepatocellular Carcinoma. Since the article compares the expertise of the surgeon as well as quality of living (before and after) surgery, it will surly improve the cancer treatment procedures positively.

Author Response

The article has been significantly improved as per the suggestions made and it will be of great help to optimize the quality of Life of patients undergone surgery for Hepatocellular Carcinoma. Since the article compares the expertise of the surgeon as well as quality of living (before and after) surgery, it will surly improve the cancer treatment procedures positively.

Ans:

Thank you for your encouraging comment.

Reviewer 2 Report (Previous Reviewer 2)

Dear Dr. Chang et al,

I revised your paper in a second round to check the modifications requested  for possible publication in Cancers as an original article.

Unfortunately your correction does not satisfy me, as the English text is very incorrect and I really ask you to fix it, especially the parts added with the latest revision, with an English native speaker (see, the example on the bottom). 

Example for modification of paragraph between lines 33 and 38.

Patients who undergone resection for HCC reach a significant improvement in QOL scores after 6 months since resection, showing then a plateau value in all subscale items during the 5-year period of the study. The QOL amelioration regards particularly patients treated with laparoscopic and robotic surgery techniques, respect to those that received open surgery, and appeared significantly better for robotic surgery.

Furthermore, you don’t have tried to graph table 2 as required. Unfortunately, the paper is not satisfactory in this way: I propose to include table 3 among the supplementary materials and to show the results of table 2 with a spider or radar diagram. There is no limitation by the editorial office, but it will be an improvement for the graphic of the paper and for understanding of readers.

Finally, I ask you to report in the text the average follow-up time of the study population and the final number of patients (i.e., evaluated at the last check-up), excluding the deceased ones into each of the 3 groups of case that received the surgical procedures described.

Author Response

I revised your paper in a second round to check the modifications requested  for possible publication in Cancers as an original article.

Unfortunately your correction does not satisfy me, as the English text is very incorrect and I really ask you to fix it, especially the parts added with the latest revision, with an English native speaker (see, the example on the bottom).

Example for modification of paragraph between lines 33 and 38.

Patients who undergone resection for HCC reach a significant improvement in QOL scores after 6 months since resection, showing then a plateau value in all subscale items during the 5-year period of the study. The QOL amelioration regards particularly patients treated with laparoscopic and robotic surgery techniques, respect to those that received open surgery, and appeared significantly better for robotic surgery.

Ans:

We have corrected all English grammar and writing errors throughout the manuscript, and the resubmitted version has been reviewed and edited by a professional technical editor who is a native English speaker. 

Furthermore, you don’t have tried to graph table 2 as required. Unfortunately, the paper is not satisfactory in this way: I propose to include table 3 among the supplementary materials and to show the results of table 2 with a spider or radar diagram. There is no limitation by the editorial office, but it will be an improvement for the graphic of the paper and for understanding of readers.

Ans:

As advised by the reviewer, the data in Table 2 are presented in radar diagrams in Figs. 2-3 in the revised manuscript. Additionally, Table 3 has been changed to Supplementary Table S1. Thank you for your suggestions.

Finally, I ask you to report in the text the average follow-up time of the study population and the final number of patients (i.e., evaluated at the last check-up), excluding the deceased ones into each of the 3 groups of case that received the surgical procedures described.

Ans:

The study flowchart has been revised as suggested by the reviewer, and relevant text in the Materials and Methods section has been revised accordingly. Again, thank you for your excellent suggestion.

Round 2

Reviewer 2 Report (Previous Reviewer 2)

Dera Authors,

Now the paper is suitable for publication on Cancers

This manuscript is a resubmission of an earlier submission. The following is a list of the peer review reports and author responses from that submission.

Round 1

Reviewer 1 Report

The article "Inverse Probability of Treatment Weighting in 5-year Quality of Life Comparison among Three Surgical Procedures for Hepatocellular Carcinoma" compares surgical procedures used for over 520 Hepatocellular Carcinoma patients to compare the effect of pre-and post- surgery Quality of Life and factors affecting them. 

Major concern:

This article explains in detail the factors affecting the quality of life of over 520 patients with Hepatocellular Carcinoma before and after surgery. Through these case studies, it seems clear that the success of their operations is affected by individual subscales of quality of life (QOL) before and after surgery.

Major Concerns:

- Line 87-88: "For accurate assessment of postoperative outcome measures, only patients who had been treated by highly experienced surgeons were analyzed" What do the authors mean to say through this study design? Must the quality of the surgeon also be considered important similar to patients' quality of life?

- Similarly, Line 89-91: "analysis was limited to patients who had received surgical resection performed by directors of surgery in a medical institution or by senior attending doctors specializing in HCC surgery or treatment.

- The 3 types of surgeries compared have varying numbers. Were they statistically normalized before deriving these conclusions?

- The different factors considered for quality of life includes the medical history of patients like Hepatitis, tumor stages, addictions, etc. How do the authors co-relate these with respect to the quality of life? Especially stages of cancer and addictions how were they monitored?

Minor Concerns:

- Living with family members has been considered an important variable. Can the author explain the ideal criteria for considering family members for this reported work? 

- Table 6 explains previously reported 5 studies carried out to study changing trends and risk factors for decreased quality of life (QOL). How this study design was different from the previous ones?

- Line 399-401: "subjects who continuously participated in the study throughout the 5 years did not significantly differ from those who died or dropped out during the observation period of the study (data not shown)." Since the data is not made available can it be considered as a limitation?

Author Response

Reviewer 1:

The article "Inverse Probability of Treatment Weighting in 5-year Quality of Life Comparison among Three Surgical Procedures for Hepatocellular Carcinoma" compares surgical procedures used for over 520 Hepatocellular Carcinoma patients to compare the effect of pre-and post- surgery Quality of Life and factors affecting them.

Major concern:

This article explains in detail the factors affecting the quality of life of over 520 patients with Hepatocellular Carcinoma before and after surgery. Through these case studies, it seems clear that the success of their operations is affected by individual subscales of quality of life (QOL) before and after surgery.

Major Concerns:

- Line 87-88: "For accurate assessment of postoperative outcome measures, only patients who had been treated by highly experienced surgeons were analyzed" What do the authors mean to say through this study design? Must the quality of the surgeon also be considered important similar to patients' quality of life?

Ans:

Thank you for your question. Previous studies have confirmed that surgeon experience may influence surgical outcomes and that the selection of a surgical procedure should consider the volume of relevant surgical procedures and interventions recently performed by the surgeon (Solomon DH, et al., Arthritis Rheum 2002;46(9):2436-2444; Katz JN, et al., Arthritis Rheum 2003;48(2):560-568). Therefore, our analysis excluded surgeons who had performed less than three procedures annually in order to focus the analysis on surgeons with the most skill and experience (lines 46-48, page 2 & lines 1-3, page 3).

- Similarly, Line 89-91: "analysis was limited to patients who had received surgical resection performed by directors of surgery in a medical institution or by senior attending doctors specializing in HCC surgery or treatment."

Ans:

To address the above concerns of the reviewer, the revised manuscript clarifies that the study was designed to focus the analysis on procedures performed by highly skilled and highly experienced surgeons (lines 46-48, page 2 & lines 1-3, page 3). Thank you.

- The 3 types of surgeries compared have varying numbers. Were they statistically normalized before deriving these conclusions?

Ans:

Previous works have already pointed out that inverse probability of treatment weighting (IPTW) is a form of PS analysis in which probability weights are used to reduce imbalance in potential confounding factors between treated and control patients (Curtis et al., Med Care 2007;45:S103–107; Austin & Stuart, Stat Med 2015;34:3661-3679; Jerome et al., JAMA Cardiol 2016;1(6):655-65 ). The following text has been added to the Statistical Analysis subsection of the revised manuscript to clarify the novel use of IPTW in this study: “A propensity score approach was applied to enable use of IPTW to balance the baseline characteristics of patients among the three surgery types. Each observation was weighted by the inverse of the probability of a patient receiving HCC surgery, given observed confounders identified to the index date. Stabilized inverse probability weights were used to mitigate the influence of very low probabilities estimated by the propensity score model [16]. Weights were derived to obtain estimates representing population-average treatment effects to enable a balanced comparison among the three groups. Treatment was considered the method chosen at the time of consent to this study. Regression models were used to make final inferences, which enabled adjustment for any covariate that remained unbalanced after IPTW” (lines 32-40, page 4). Thank you. 

- The different factors considered for quality of life includes the medical history of patients like Hepatitis, tumor stages, addictions, etc. How do the authors co-relate these with respect to the quality of life? Especially stages of cancer and addictions how were they monitored?

Ans:

The revised Statistical Analysis section clarifies how the QOL estimates in our study accounted for such factors (lines 42-51, page 4). Thank you.

Minor Concerns:

- Living with family members has been considered an important variable. Can the author explain the ideal criteria for considering family members for this reported work?

Ans:

Our ideal criteria for considering the role of family members in this study were the availability of relevant medical records and the ability to collect baseline data through in-person interviews. Thank you.

- Table 6 explains previously reported 5 studies carried out to study changing trends and risk factors for decreased quality of life (QOL). How this study design was different from the previous ones?

Ans:

The resubmitted version distinguishes our study from previous works by highlighting our use of a larger cohort population and more numerous longitudinal time points compared with the selected studies (lines 46-50, page 12). Thank you.

- Line 399-401: "subjects who continuously participated in the study throughout the 5 years did not significantly differ from those who died or dropped out during the observation period of the study (data not shown)." Since the data is not made available can it be considered as a limitation?

Ans:

The above sentence has been moved to the Results section in the revised manuscript (lines 25-27, page 5). Thank you.

Reviewer 2 Report

Dear Authors,

The QOL analysis by Chang et al. in cases with hepatocellular carcinoma treated with different resective surgery techniques (robotic, VLS and open) is a study that includes 520 cases, followed with an average greater than 4 years, compared after IPTW matching, and assessed at subsequent time points (at baseline, 6 months and 3 and 5 years) with 2 scales (FACT-Hep and EORTC).

The study appears well presented, clear in the methodology used and discussed in comparison with other studies reported in the literature in an organic and critical way.

Unfortunately, the data are reported with too many tables, which can bore the reader and I would ask the Authors to consider a graphical presentation of the results, for example with radar or spider diagram, particularly of table 2 data.

I suggest to move table 3 as supplementary material, removing it from the paper and adding the subscales (i.e., comparison T2 vs T1 and so on ...) in the table that are missing.

In table 1, I ask if it is possible to report the number of cases also in the columns of the section after IPTW matching.

Tables 4 and 5 have incorrect captions, please correct them.

Author Response

Reviewer 2:

Dear Authors,

The QOL analysis by Chang et al. in cases with hepatocellular carcinoma treated with different resective surgery techniques (robotic, VLS and open) is a study that includes 520 cases, followed with an average greater than 4 years, compared after IPTW matching, and assessed at subsequent time points (at baseline, 6 months and 3 and 5 years) with 2 scales (FACT-Hep and EORTC).

The study appears well presented, clear in the methodology used and discussed in comparison with other studies reported in the literature in an organic and critical way.

Ans:

Thank you for your encouraging comment.

Unfortunately, the data are reported with too many tables, which can bore the reader and I would ask the Authors to consider a graphical presentation of the results, for example with radar or spider diagram, particularly of table 2 data.

Ans:

Thanks for your valuable comments. We agree that the use of a radar or spider diagram would be more appealing to readers. Unfortunately, our presentation of data for this manuscript was limited by the journal constraints on word and page counts.  Therefore, we will use alternative graphical presentations of our data in our next manuscript.

I suggest to move table 3 as supplementary material, removing it from the paper and adding the subscales (i.e., comparison T2 vs T1 and so on ...) in the table that are missing.

Ans:

As advised by the reviewer, Table 3 has been changed to Supplementary Table S1, and the missing subscales have been added and labeled accordingly. Thank you.

In table 1, I ask if it is possible to report the number of cases also in the columns of the section after IPTW matching.

Ans:

As advised by the reviewer, the revised Table 1 reports case number after IPTW matching. Thank you for your suggestion. 

Tables 4 and 5 have incorrect captions, please correct them.

Ans:

The captions for Tables 4-5 have been corrected. Thank you.
